# Last Decade Progress in Understanding and Modeling the Land Surface Processes on the Tibetan Plateau

Hui Lu[1, 4], Donghai Zheng[2], Kun Yang[1, 3], Fan Yang[1]

[1]Ministry of Education Key Laboratory for Earth System Modeling, Department of Earth System Science, Tsinghua University, Beijing 100084, China
[2]National Tibetan Plateau Data Center, Key Laboratory of Tibetan Environmental Changes and Land Surface Processes, Institute of Tibetan Plateau Research, Chinese Academy of Sciences, Beijing 100101, China
[3]Center for Excellence in Tibetan Plateau Earth Sciences, Chinese Academy of Sciences, Beijing 100101, China
[4]Ministry of Education Ecological Field Station for East Asian Migratory Birds, Beijing 100084, China

*Correspondence to*: Hui Lu (luhui@tsinghua.edu.cn)

**Abstract.** The Land Surface Model (LSM) that simulates water and energy exchanges at the land-atmosphere interface is a key component of the Earth system model. The Tibetan Plateau (TP) drives the Asian monsoon through surface heating and thus plays a key role in regulating the climate system in the Northern Hemisphere. Therefore, it's vital to understand and represent well the land surface processes on the TP. After an early review that identified key issues in the understanding and modelling of land surface processes on the TP in 2009, several progresses have been made in the last decade in developing new land surface schemes and supporting datasets. This review summarizes the major advances, including (i) An enthalpy-based approach was adopted to enhance the description of cryosphere processes such as glacier/snow mass balance and soil freeze-thaw transition. (ii) Parameterization of the vertical mixing process was improved in lake models to ensure reasonable heat transfer to the deep water and to the near-surface atmosphere. (iii) New schemes were proposed for modelling water flow and heat transfer in soils accounting for the effects of vertical soil heterogeneity due to the presence of high soil organic matter content and dense vegetation roots in surface soils, or gravel in soil columns. (iv) Supporting datasets of meteorological forcing and soil parameters were developed by integrating multi-source datasets including ground-based observations. Perspectives on the further improvement of land surface modelling on the TP are provided, including the continuous updating of supporting datasets, parameter estimation through assimilation of satellite observations, improvement of snow and lake processes, adoption of data-driven and artificial intelligence methods, and the development of an integrated LSM for the TP.

## 1 Introduction

The Land surface model (LSM) is a key component of the earth system model, which simulates the water and energy exchanges at the land–atmosphere interface (Dickinson et al., 2006; Hurrell et al., 2013). Moreover, the LSM is a powerful tool to provide a better understanding of the interactions among hydrological, ecological, and biogeochemical processes (Pan et al., 2012). Over the past decades, the LSMs have evolved from a simple bucket model (Manabe, 1969) to more sophisticated model systems (Dai et al., 2003; Dickinson et al., 1993; Oleson et al., 2010; Sellers et al., 1986; Sellers et al., 1996) that have

incorporated many physical and physiological processes occurring in the atmosphere-snow-vegetation-soil-aquifer system. Meanwhile, several offline multi-model intercomparison projects have been conducted to identify the strength and weakness of LSMs, such as the Project for the Intercomparison of Land-Surface Parameterization Schemes (PILPS) (Henderson-Sellers et al., 1995; Henderson-Sellers et al., 1993), the Global Soil Wetness Project (GSWP) (Dirmeyer et al., 1999; Dirmeyer, 2011), and the African Monsoon Multidisciplinary Analyses (AMMA) Land Surface Model Intercomparison Project (ALMIP) (Boone et al., 2009). A general conclusion is that there are still big differences among various LSMs and none of them outperforms others for all conditions. Jimenez et al. (2011) and Xia et al. (2014) also reported that large discrepancies exist between various LSM simulations even when driven with the same meteorological forcing. Therefore, further improvement of LSMs remains imperative.

The Tibetan Plateau (TP), well-known as the Third Pole and the Asian water tower, shows very strong interactions among its hydrosphere, cryosphere, biosphere, and atmosphere that manifest TP's impact on surrounding regions and regional/global climate system (Wu et al., 2015; Yao et al., 2015). The land surface processes in the TP show thus unique features due to the multispherical interactions. The land-atmospheric exchange on the TP is generally intense due to strong solar radiation, which is further considerably regulated by the complex terrain. Typical processes of the cryosphere (such as glacier, soil freeze-thaw, and snow processes), lake-air interactions, and vertical soil heterogeneity coexist and consist of the TP's complex land surface processes that are not well represented by current LSMs. The coupled land-atmospheric numerical experiments (Gao et al., 2015; Zhao et al., 2018) demonstrate also the important role of the TP's land surface processes on affecting the simulations of atmospheric processes and variables. These numerical experiments further highlight the necessity to improve current LSMs to better represent the TP's complex land surface processes.

After Yang et al. (2009) identified key issues in the understanding and modelling of land surface processes on the TP, many endeavors have been made to evaluate and validate the performance of current LSMs, as well as to identify the model deficiencies and to propose solutions (Wang et al., 2016b; Zheng et al., 2017b; Zheng et al., 2016). Based on the evaluation results, the LSMs have been improved from two aspects. One is to improve the existing schemes, such as improvement of parameter values (Dai et al., 2013; Shangguan et al., 2014; Shangguan et al., 2013; Shangguan et al., 2017), development of more realistic and reliable process representations (Wang and Yang, 2018; Wang et al., 2020a), and usage of better mathematic solutions (Wang et al., 2017; Luo et al., 2018). The other is to extend modelling capability through adding new schemes and functions that are missing or ignored in current LSMs (Chen et al., 2012; Zheng et al., 2015a, b; Zhu et al., 2017). In addition, new meteorological forcing datasets have also been developed by integrating multi-source datasets including ground-based observations (He et al., 2020).

To the best of our knowledge, there is no systematic review of the last decade's progress in understanding and modelling the land surface processes on the TP. The presented topics in this paper cover the new development of land surface schemes for the TP (section 2) and the supporting datasets such as meteorological forcing and soil datasets (section 3). Finally, perspectives on the further improvement of land surface modelling on the TP are given (section 4).

## 2 Advances of land surface modelling on the TP

### 2.1 Cryosphere processes

Deficiencies in representing soil freeze-thaw and glacier/snow processes in cold and high mountainous regions are the main sources of uncertainties for applying the LSMs to the TP's cryosphere. Land surface processes in the cryosphere, including the phase change of water and water-heat interactions, are more complex than those in unfrozen areas. Zheng et al. (2017b) showed that the rates of water phase change are different in freezing and thawing periods, and the concept of phase change efficiency is introduced for improving the simulation of soil freeze-thaw processes as confirmed by Yang et al. (2018). A major advance in representing the transition process of water phase change on the TP is the introduction of enthalpy in the energy governing equations for heat transfer. The introduction of enthalpy not only makes the governing equation simpler but also makes the solution more stable. Enthalpy was used as a prognostic variable instead of temperature to simulate snow accumulation and melting (Sun et al., 1999), lake ice formation and melting (Sun et al., 2007), as well as soil freeze-thaw processes (Li and Sun, 2008). Based on these pioneer practices, Bao et al. (2016) developed a frozen soil scheme, and Wang et al. (2017) further coupled this scheme with a snow scheme originally from Sun et al. (1999), in which enthalpy was adopted as a prognostic variable instead of snow/soil temperature for simulating heat transfer. The coupled model of Wang et al. (2017) was proved to be able to realistically capture the measured soil/snow water and temperature dynamics on the TP, confirming the advantage of adopting the enthalpy as the prognostic variable for snow and frozen soil simulations.

The enthalpy concept was also applied to the development of a glacier mass balance model, i.e. the Water and Enthalpy Budget-based Glacier mass balance Model (WEB-GM) (Ding et al., 2017). The WEB-GM contains a dynamic scheme to judge precipitation type (rain/sleet/snow) based on wet-bulb temperature, relative humidity, and elevation (Ding et al., 2014), a new albedo scheme to consider the impacts of shallow snow, and a parameterization scheme for turbulent heat transfer (Yang et al., 2008). Compared to the traditional temperature-based glacier mass balance model (i.e. Fujita model), the WEB-GM model performs better in simulating glacier surface temperature, turbulent heat fluxes and glacier mass balance at the Parlung No.4 Glacier in the southeast TP (Fig. 1).

Since soil water flow and heat transport are strongly coupled in the freeze-thaw transition processes, Wang and Yang (2018) incorporated a fully coupled soil water-heat transport scheme into the CLM4.5 model, whereby the interaction between soil water flow and heat transport as well as the soil water vapor diffusion process are considered. The results show that the measured frozen soil dynamics (e.g. soil moisture and temperature) in the TP are better captured in comparison to the original isothermal scheme, which also highlight the importance of considering the process of water vapor diffusion in cold regions that is generally neglected in current LSMs. Considering that the snow cover on the TP is generally thin (~ a few centimeters) and therefore the subsurface albedo has a large influence on the snow surface albedo, a new snow albedo scheme was developed recently, with which simulations of snow ablation were improved (Wang et al., 2020a).

## 2.2 Lake processes

In recent years, modelling of lake-air interactions has drawn more and more attention, and hydro-meteorological models such as WRF and CLM have included lake models as a component. The WRF-lake model generally performs well for shallow lakes (Gu et al., 2015). However, Fang et al. (2017) showed that the WRF-lake model performs poorly when applied to Lake Nam

Co, a deep lake, on the TP. Compared to the in situ measurements, the simulated lake surface temperature is too low, the lake ice appears too early, and the seasonality of lake temperature in deep water is too smooth. These biases indicate that the WRF-lake model underestimates the process of strong vertical mixing between surface water and deep water. On the other hand, Lazhu et al. (2016) found that a semi-empirical lake model, Flake, can reproduce well the measurements of surface temperature, surface turbulent fluxes and ice phenology in the Lake Nam Co. The performance evaluations of the above two models indicate

that a semi-empirical model like Flake may be superior for the application to deep lakes on the TP. The Flake simulations also showed that the actual lake evaporation is less than the potential evaporation that is traditionally used as a surrogate of the former (Lazhu et al., 2016). Huang et al. (2019) also compared the performance of Flake, WRF-lake and CoLM-lake models for their applications to Lake Nam Co, which further investigated the sensitivity of lake models to several key parameters, such as light extinction coefficient, turbulent mixing, and temperature of maximum water density. The results demonstrated

that the simulations can be largely improved via model calibrations.

Based on the above model evaluations, lake models have been improved through better representing the processes of lake-air interaction and lake interior vertical mixing on the TP. Based on in situ measurements, Wang et al. (2015) evaluated the performance of two lake-air exchange models in the small Nam Co lake and showed that the simulated surface heat fluxes can be enhanced by increasing the values of Charnock and roughness Reynold numbers from 0.013 and 0.11 to 0.031 and 0.54,

respectively. Zhang et al. (2019) developed a new scheme to improve vertical mixing in the lake interior based on a K-profile parameterization (KPP) associated with internal wave activity and shear instability. The KPP was incorporated into the CLM lake model (CLM-KPP), which represents better the transition between stratification and turnover and effectively reproduces strong mixing between the lake surface and deep water during windy conditions, leading to improved simulation of water surface temperature in comparison to the original model (CLM-ORG) (Fig. 2).

In addition to the improvements in those general processes, lake models were developed to consider some typical processes for the TP lakes. Dai et al. (2018) developed the CoLM-Lake with a focus on freezing and thawing transition of lakes, which was tested at 10 lakes worldwide, and the results show that the model can perform well if model parameters are calibrated. Considering that there are a number of salt lakes in the TP, Su et al. (2019) incorporated the salinity effects on the temperatures of maximum water density and freezing point into the Flake model and validated its performance in the brackish endorheic

Qinghai Lake. The results showed that consideration of salinity effects reduces the simulated maximum ice thickness and advances the break-up date that are closer to the in situ buoy measurements.

## 2.3 Soil subsurface processes

Soil subsurface processes in the TP are largely affected by the vertical heterogeneity. The vertical stratification is caused by the presence of high soil organic matter (SOM) content and dense vegetation roots in surface soils (Yang et al., 2005; Zheng et al., 2015a, b) or gravel (Yi et al., 2018) in soil columns. Their existences considerably influence soil thermal and hydraulic properties and therefore soil state and land surface fluxes.

The existence of SOM in the surface soils leads to lower thermal conductivity and larger heat capacity (Zheng et al., 2015b; Chen et al., 2012). Chen et al. (2012) evaluated three widely used thermal conductivity schemes (e.g. Farouki, 1981; Johansen, 1975; Yang et al., 2005) implemented by current LSMs using laboratory measurements. The results showed that all three schemes overestimate the measured thermal conductivity for Tibetan soils including SOM content, and the scheme of Yang et al. (2005) was further modified to include the SOM effect that reduces the noted overestimations (Fig. 3). Similar work was done by Zheng et al. (2015b) to incorporate the SOM effect into the scheme of Johansen (1975). Luo et al. (2017) showed that consideration of SOM effect in the thermal scheme can also improve the soil temperature simulations produced by the CLM4.5 model for two Tibetan grassland sites. Zheng et al. (2015b) further demonstrated that consideration of SOM effect tends to produce larger sensible heat flux and smaller latent heat flux, and the amplitude of the diurnal surface temperature cycle is increased.

The existence of SOM in the surface soils also increases soil porosity and water holding capacity (Chen et al., 2012; Zheng et al., 2015a). Previous studies (Chen et al., 2013; Li et al., 2018a; Li et al., 2017; Su et al., 2013) have shown that current LSMs generally underestimate the surface soil moisture in the TP by ignoring the SOM effect in the hydraulic parameterization. Chen et al. (2012) and Zheng et al. (2015a) showed that the presence of SOM content in the surface layer of Tibetan soils leads to larger porosity and higher water holding capacity, and the saturated hydraulic conductivity is found exponentially decreased with soil depth (Zheng et al., 2015a). Accordingly, a new hydraulic parameterization was developed by Zheng et al. (2015a) to include the above SOM effect, which reduces the noted underestimation of surface soil moisture under wet conditions. However, an overestimation of surface soil moisture was noted for the drying period, which can be addressed by implementing either the exponential or asymptotic vertical root distribution function that is able to represent the presence of abundant roots in the surface layer (Zheng et al., 2015a; Zheng et al., 2015c). Besides affecting the water uptake, Gao et al. (2015) inferred that the presence of the rhizosphere in the surface layer could reduce the saturated hydraulic conductivity and infiltration that increases the water holding capacity, which can also mitigate the dry biases noted for the simulation of surface soil moisture on the TP.

The existence of gravels is another characteristic that should be considered in some part of the TP (Yi et al., 2018), in which soils are usually not well-developed and large gravels (particle size ≥ 2 mm) may occupy considerable volumes. The existence of gravels tends to decrease the water holding capacity and increase the hydraulic conductivity and thermal conductivity. Laboratory experiments were conducted by Yi et al. (2018) that for the first time the impact of gravels on soil hydraulic and thermal parameters as well as the soil water content and frozen soil depth was quantified. A new soil hydraulic and thermal

scheme was also developed by Pan et al. (2017) for simulating the freeze-thaw process in gravel-contained soils. The scheme was incorporated into the CLM4.0 model that captures well the measured freeze-thaw dynamics in two typical Tibetan sites that contain different levels of gravel content in the soil columns.

## 2.4 Hydrological processes

The understanding and modelling of hydrological characteristics on the TP, such as the runoff seasonality and parameterization
of subsurface/groundwater flow, was also much improved. Bai et al. (2016) evaluated the performance of GLDAS products in representing the streamflow dynamics of five Tibetan river basins based on the gauged measurements. It demonstrated that the four LSMs (CLM, VIC, Noah, and MOSAIC) implemented by the GLDAS capture well the trend of annual streamflow dynamics, but fail to reproduce the seasonal pattern such as the timing of peak flow. Zheng et al. (2018; 2016) showed that the seasonality of the runoff regime in the eastern Tibetan river basins is mainly controlled by the soil freeze-thaw mechanism
(Fig. 4a-4b). An annual hysteresis loop between measured/simulated runoff and precipitation was also reported (Fig. 4c-4d), which is related to both the amount and the state of water stored in the soils. Zheng et al. (2017a) further showed that the soil water storage-based runoff parameterizations outperform the groundwater table-based schemes in representing the runoff dynamics in the seasonally frozen Tibetan river basins as confirmed by Yuan et al. (2018). Both Bai et al. (2016) and Zheng et al. (2017a) also highlight the importance of including groundwater and lateral flow schemes in current LSMs. Recently, a
groundwater scheme (Xie et al., 2012; Zeng et al., 2016a; Zeng et al., 2016b; Xie et al., 2018) has been developed and coupled into the CLM4.5 model, and the results demonstrated that the scheme can simulate well the groundwater level and lateral flow. Nevertheless, it is still a big challenge to simulate well the hydrological processes across all the Tibetan rivers due to the lack of gauged measurements (including precipitation and streamflow) in most parts of the TP.

## 2.5 Representation of sub-grid topographic effects

The TP is characterized by high elevation and complex terrain, which exert obvious sub-grid topographic effects on radiative and moment transfer processes. A 3D Monte Carlo photon tracing program for radiation transfer was developed by Liou et al. (2007) and applied to simulate the surface solar fluxes on the TP. It was found that the inclusion of subgrid variability can lead to a significant solar flux deviation from the conventional smoothed topography simulation. Lee et al. (2013) further improved this 3D model and applied it to investigate the topographic effect on surface radiative energy budget on the TP. The results
demonstrated that the TP would receive more solar flux that leads to stronger convection and enhanced snowmelt rate when the 3D topographic effects were considered. Recently, Lee et al. (2019) investigated the climate effects of the 3D radiation-topography interactions by incorporating their model into the Community Climate System Model version 4 (CCSM4). After including this topographic effect, the energy budget and air temperature simulated by the CCSM4 could be significantly improved. Gu et al. (2020) also coupled a subgrid terrain radiative forcing scheme into the Regional Climate Model Version
4.1 (RegCM4.1). They found that adopting this scheme in the RegCM4.1 could produce a better simulation of the energy budget on the TP, which thus lead to a better simulation of East Asian summer monsoon and precipitation over China.

Representing the effect of subgrid terrain variability on the moment transfer is very important for improving the simulation of wind speed, water vapour transfer, and precipitation over the TP. Wang et al. (2017) demonstrated that water vapour transfer over the South Tibetan Plateau is overestimated by current climate models. Lin et al. (2018) also found similar biases in WRF simulations and demonstrated that the complex terrain of the Himalaya can retard water vapour transfer through exerting an orographic drag on the atmosphere. Han et al. (2015) suggested an aerodynamic length of about 10 m is needed to reflect the drag, which, however, is difficult to apply such a "big number" directly to the framework of Monin-Obhukov similarity theory. Alternatively, Zhou et al. (2018; 2019) implemented a turbulent orographic form drag scheme (Bejaars et al., 2004) in the WRF model, which reduce the positive biases in the surface wind simulation, leading to the reduction of water vapour transfer from the surroundings to the TP. This parameterization is also critical for the successful simulation of precipitation distribution from the low altitudes to high latitudes of the central Himalaya (Wang et al., 2020).

## 3 Improvement of supporting data

Offline simulation is generally adopted to test model performance, and meteorological forcing, as well as static land parameter datasets, are required for running the LSM. There are several global forcing datasets dedicated to driving the LSMs, such as the global land data assimilation system (GLDAS) (Rodell et al., 2004), ERA-Interim (Dee et al., 2011), and Modern-Era Retrospective Analysis for Research and Applications (MERRA) (Gelaro et al., 2017; Rienecker et al., 2011). Wang and Zeng (2012) compared the performance of several precipitation datasets derived from GLDAS, MERRA, NCEP/NCAR-1, CFSR, ERA-40, and ERA-Interim for their applications to the TP. The results showed that the GLDAS performs better than other datasets due to the fact that the GLDAS has merged products from the ground- and satellite-based measurements. However, the development of these datasets only utilized measurements from limited Chinese weather stations (~200 stations), and their performance can be improved if more ground-based measurements are used (Wang and Zeng, 2012; Wang et al., 2016a; Wang et al., 2016c). On the other hand, the global land parameter datasets such as the soil data derived from the FAO Soil Map (FAO, 2003), Harmonized World Soil Database (HWSD) (FAO/IIASA/ISRIC/ISS-CAS/JRC, 2012), and SoilGrid (Hengl et al., 2014) show similar problems. Accordingly, several meteorological forcing and soil datasets have been developed specifically for the mainland of China via merging more available ground-based measurements during the last decade. These improved forcing data and other supporting data generally outperform the corresponding global data set not only in accuracy but also in both spatial and temporal resolution, which make a finer scale land surface simulation possible and contribute to substantial improvements of land surface modelling in China.

## 3.1 Improvement of meteorological forcing data

The accuracy of LSM simulations highly depends on the quality of meteorological forcing data. Among them, the accuracy of precipitation and radiation data is more addressed. For lake simulations, particularly for modeling lake ice, it is also crucial to provide accurate information on wind speed.

Table 1 lists several commonly-used precipitation datasets in China. Among them, the CN05.1 was produced by China Meteorological Administrator through the interpolation from more than 2000 gauge stations over the mainland of China (Wu and Gao, 2013). The CMFD was developed by the Institute of Tibetan Plateau Research, Chinese Academy of Sciences (ITPCAS) via merging a variety of data sources including reanalysis data, satellite-based products, and ground-based measurements (He et al., 2020). Fig. 5 shows the comparison of mean annual precipitation derived from the six forcing datasets for the TP during the period between 1985 and 2014. Large differences can be noted among the six precipitation datasets. According to the CN05.1 that produced based on the interpolation of ground-based measurements, the mean annual precipitation is about 410 mm on the TP with a larger amount in the southeast and smaller amount in the northwest. Compared to the CN05.1, both the ERA-Interim and MERRA2 largely overestimates the precipitation in the southeastern part of the Himalaya, and the precipitation in the southeastern part of the TP is generally overestimated by the MERRA2. On the contrary, the GLDAS-1 underestimates the precipitation in the southeastern part of the TP, which is resolved in the GLDAS-2. However, the overestimation of precipitation in the central part of Himalaya can be noted for the GLDAS-2. Overall, the CMFD integrating the ground-based measurements produce spatial distribution and variation of precipitation comparable to the CN05.1. The reliability of the CMFD precipitation data has also been validated by other studies (He et al., 2015; Xie et al., 2017; Yang et al., 2017). Yang et al. (2017) evaluated the precipitation data of the CMFD, CN05.1, and GLDAS using independent ground-based measurements collected by the China Ministry of Water Resources (MWR). The results demonstrated that the CMFD generally outperforms the CN05.1 and GLDAS in capturing the spatial and temporal variability of precipitation in China. Similar findings were also reported by He et al. (2015) for their applications to the upper reach of Heihe River Basin.

Until now, the CMFD forcing data has been adopted in hundreds of research publications with respect to land surface simulations, ecological modeling, and climate change studies. For instance, the data was used to (i) improve the land surface temperature simulation in China (Chen et al., 2011), (ii) simulate the distribution of permafrost and seasonal frozen soil on the TP (Guo and Wang, 2013), (iii) drive the LSM to obtain a high spatial resolution simulation of soil moisture on the TP (Li et al., 2018a), and (iv) drive a land data assimilation system to improve the soil moisture simulation in China (Yang et al., 2020; Yang et al., 2016).

### 3.2 Improvement of static land parameter data

Recently, a series of land parameter datasets have been developed for running the LSMs by the Land-Atmosphere Interaction Research Group at Beijing Normal University and Sun Yat-sen University available from http://globalchange.bnu.edu.cn/research/data. These datasets include (i) a soil particle-size distribution dataset of China (Shangguan et al., 2012), (ii) a China dataset of soil properties (Shangguan et al., 2013), (iii) a China dataset of soil hydraulic parameters (Dai et al., 2013), (iv) a Global Soil Dataset for use in Earth System Models (GSDE) (Shangguan et al., 2014), (v) a global depth to bedrock dataset (Shangguan et al., 2017), (vi) a global dataset of soil hydraulic and thermal parameters (Dai et al., 2019), and (vii) a reprocessed global LAI dataset (Yuan et al., 2011). Among these datasets, the GSDE has been widely

used, which provides soil information including soil texture, organic carbon, and bulk density at a spatial resolution of 30 arc-seconds for eight vertical soil layers to a depth of 2.3 m.

Fig. 6 and Fig. 7 show respectively the spatial distribution of bulk density and clay content on the TP derived from the GSDE and two other widely-used soil datasets, i.e. the HWSD and SoilGrid provided by the ISRIC (International Soil Reference and
Information Centre). Large differences can be also noted among the three soil datasets. Compared to the GSDE that produced based on more soil samples taken from China, the SoilGrid tends to provide larger values in the northwest and smaller values in the southeast for the bulk density, and the HWSD tends to provide lower values of clay content. Since soil hydraulic parameters are closely related to the bulk density and clay content, different values of bulk density and clay content derived from different soil datasets would affect the soil hydraulic properties and thus the water budget simulation by the LSMs. For
instance, Li et al. (2018b) investigated the impact of soil texture on the LSM simulations over the central Tibetan Plateau, in which the FAO Soil Map and GSDE were used. The results showed that soil parameters play a dominative role in simulating energy and water fluxes. Therefore, additional efforts are needed to investigate the accuracy of these newly developed soil datasets, as well as to investigate their impacts on the LSM simulations.

Besides the abovementioned improved forcing and soil data sets, other remote sensing or reanalysis data sets have also been
widely used for land surface modeling on the TP, especially in ungauged/poor-gauged areas. They were used to drive hydrological simulation (Wang et al., 2015; Tong et al., 2014; Qi et al., 2018) or land data assimilation system (Lu et al., 2012), to validate model simulation (Li et al., 2020), and to improve understanding of physical processes (Liu et al., 2018). On the other hand, current in situ observation networks also made contributions to understanding the land surface process by serving as calibration and validation sources. For example, the multiscale soil moisture and freeze–thaw monitoring network on the
central TP (CTP-SMTMN) (Yang et al., 2013) and the Tibetan Plateau observatory of plateau scale soil moisture and soil temperature (Tibet-Obs) (Su et al., 2011) provide valuable data set to validate and calibrate remote sensing products and model simulations. However, the density of in situ observation networks on the TP is still very sparse. Consequently, integration of the ground observation, remote sensing products, and reanalysis data may provide a complete and reliable supporting data set for studying the land surface processes on the TP.

**4 Summary and perspectives**

After Yang et al. (2009) identified key issues in the understanding and modelling of land surface processes on the TP, several progress has been made in developing new land surface schemes and supporting datasets in the past decade. The main advances are summarized in this mini-review, including (i) an enthalpy-based approach was adopted to enhance the description of cryosphere processes such as glacier/snow mass balance and soil freeze-thaw transitions; (ii) parameterization of vertical
mixing process was improved in lake models to ensure reasonable heat transfer into deep water; (iii) new schemes were proposed for modelling water flow and heat transfer in the soils accounting for the effect of vertical soil heterogeneity due to the presence of high soil organic matter content and dense vegetation roots in surface soils or gravel in soil columns; (iv)

supporting datasets of meteorological forcing and soil parameter datasets were developed by integrating more ground-based measurements.

Based on the review, the following research perspectives in land surface modeling on the TP are given.

(1) Continuous improvement of meteorological forcing and land parameter datasets. The quality of current datasets are generally good for East China but not so reliable for the Tibetan Plateau and Northwest China, because of limited access to station data. With the progress in data sharing policy, it is possible to merge more station data from these remote regions. In addition, new experimental activities (e.g. the Second Scientific Expedition to the Tibetan Plateau and the Third Atmospheric

Science Experiment on the Tibetan Plateau) may provide new observations to improve/validate the quality of the forcing datasets. Also, high-resolution modeling can provide a new meteorological data source to improve the data quality in regions with complex terrain (Li et al., 2020; Wang et al., 2020b).

(2) Estimation of land parameters at model grid-scale through data assimilation and/or multi-objective optimization. Parameter upscaling faces big challenges with field sampling. By contrast, satellite data can avoid this kind of issues, although the remote

sensing products may also encounter uncertainties. Satellite data assimilation and/or multi-objective optimization may provide a new feasible way to estimate model parameters. A recent work (Luo et al., 2020) proves the concept, showing inverse estimation with in situ data is indeed able to reproduce the spatial pattern of soil organic carbon content measured in a mesoscale area of the TP and some pioneering studies (Corbari et al., 2015; Pinnington et al., 2018; Yang et al., 2016) show the potential of this type of methods.

(3) Development of new schemes and integration of multiple schemes into a single LSM. Some parameterizations are still questionable or even excluded, such as rhizosphere effect, complex terrain, and high elevation lakes, which are typical for the TP. For example, the parameterization of the light extinction coefficient and turbulent mixing in lakes are critical to freeze-up simulation but their values are empirically given (Huang et al., 2019). Meanwhile, since the vegetation types on the TP have unique characteristics and are very sensitive to climate change, the parameterization of vegetation-related processes should be

improved as demonstrated by Li et al (2019). The role of these processes should be understood and parameterized with new observations. Coupling new schemes into a LSM can increase the diversity of the LSM community and thus better quantify the uncertainties of LSMs.

(4) Adoption of data-driven methods and artificial intelligence in land surface process study.  Big data methods could provide vast data and clues in model development, parameter optimization, and so on (Guo, 2017), while artificial intelligence could

help in process understanding (Reichstein et al., 2019). As demonstrated by Jing et al. (2020), the integration of physical process-based models with data-driven approaches shows high potential to improve the understanding and simulation of land surface processes.

**Data availability.**

Sources of the precipitation data and soil data have been described in section 3, "Improvement of supporting data".

**Author contributions.**

HL and KY conceptualized the work. HL drafted the paper. DZ and KY revised the paper. HL, DZ, KY and FY collected data and contributed to analyses and discussions.

**Competing interests.**

The authors declare that they have no conflict of interest.

**Acknowledgements**

This work was jointly supported by the Second Tibetan Plateau Scientific Expedition and Research Program (2019QZKK0206), the National Key Research and Development Program of China (2017YFA0603703), the Strategic Priority Research Program of Chinese Academy of Sciences (XDA20100103), and the National Natural Science Foundation of China (91747101).

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

**Table 1. List of six commonly-used precipitation data**

| Precipitation DataSets | Available period | Spatial resolution | Producer |
| --- | --- | --- | --- |
| CN05.1 | 1961—2014 | 0.25°×0.25° | CMA |
| CMFD | 1979 —2018 | 0.1°×0.1° | ITPCAS |
| ERA-Interim | 1979—2019 | 0.125°×0.125° | ECMWF |
| MERRA2 | 1979—2016 | 0.5° × 0.66° | NASA |
| GLDAS-1 | 1979—2019 | 1°×1° | NASA |
| GLDAS-2 | 1948—2014 | 0.25°×0.25° | NASA |

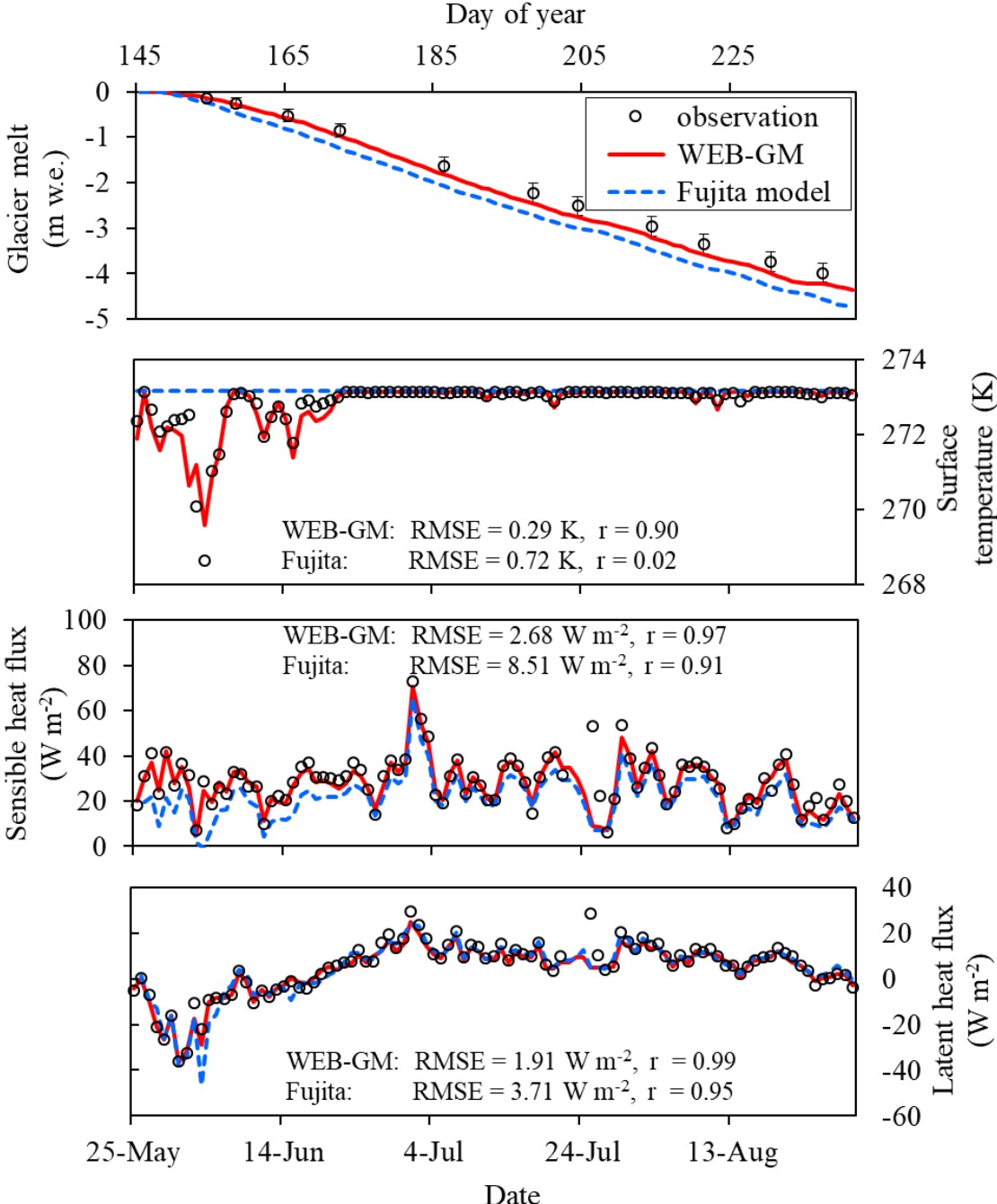

**Fig. 1: Comparisons of the WEB-GM and Fujita model in simulating glacier melt (a), surface temperature (b), sensible (c) and latent heat fluxes (d) at the Parlung No. 4 Glacier. The figure is reproduced from Ding et al. (2017).**

## (a) Water surface temperature

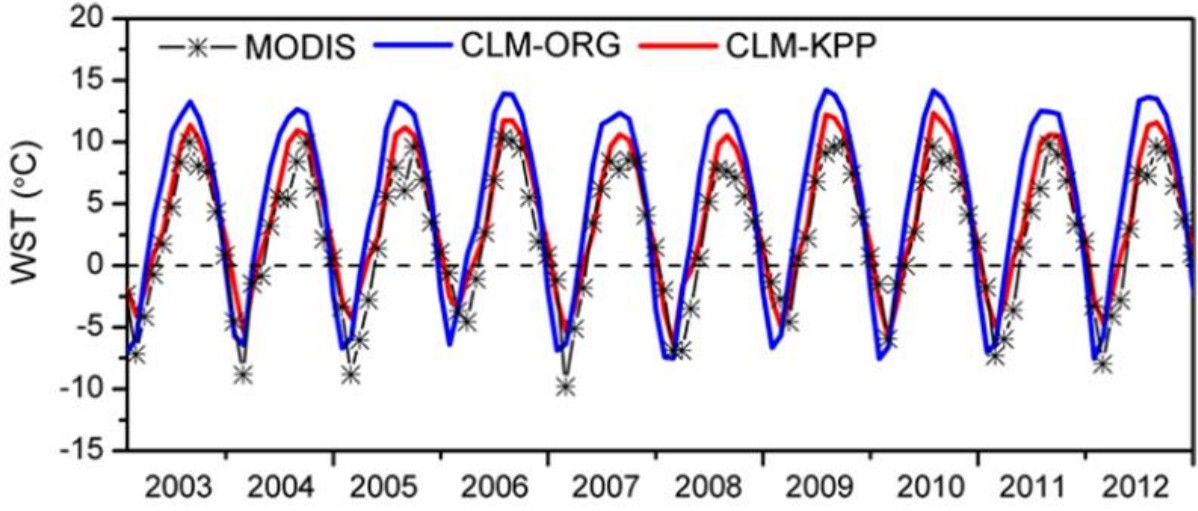

## (b) Difference between simulated water diffusivity

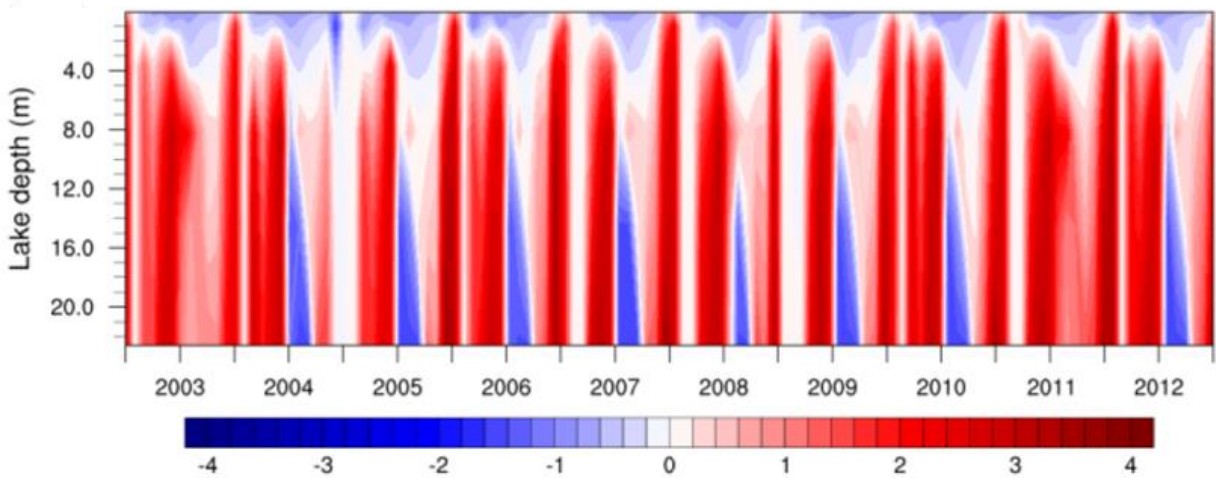

**Fig. 2: Comparisons of the original (CLM-ORG) and revised (CLM-KPP) CLM lake models in simulating water surface temperature (WST) (a), and their differences in simulating water diffusivity (log10Kw) (b) in Nam Co lake. The figure is reproduced from Zhang et al. (2019).**

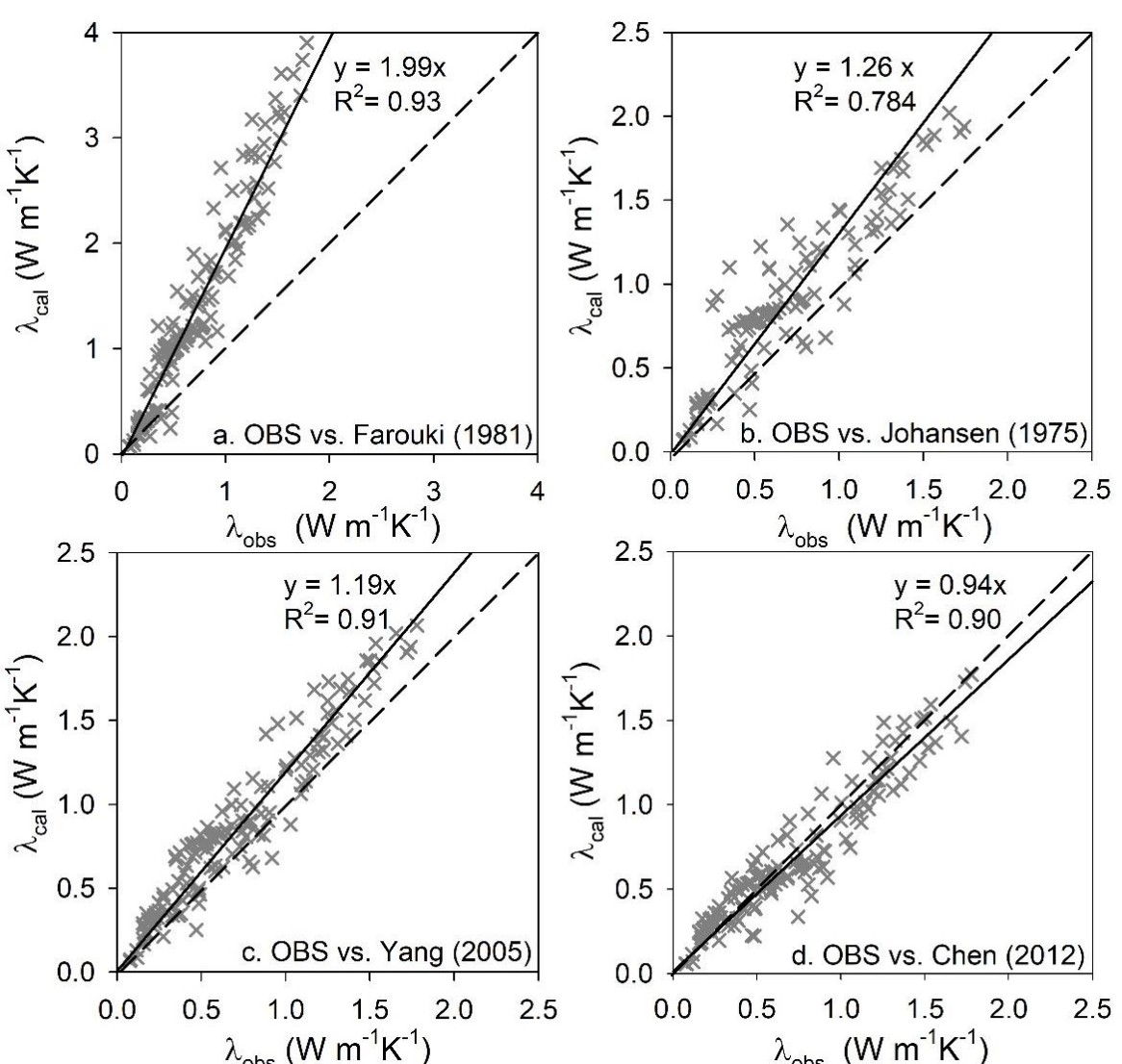

**Fig. 3: Scatterplots of laboratory-measured (OBS, shown in x-axis) and estimated thermal conductivity produced by the schemes of**
**Farouki (1981) (a), Johansen (1975) (b), Yang et al. (2005) (c), and Chen et al. (2012) (d) for 4 alpine grassland stations. The figure**
**is reproduced from Chen et al. (2012).**

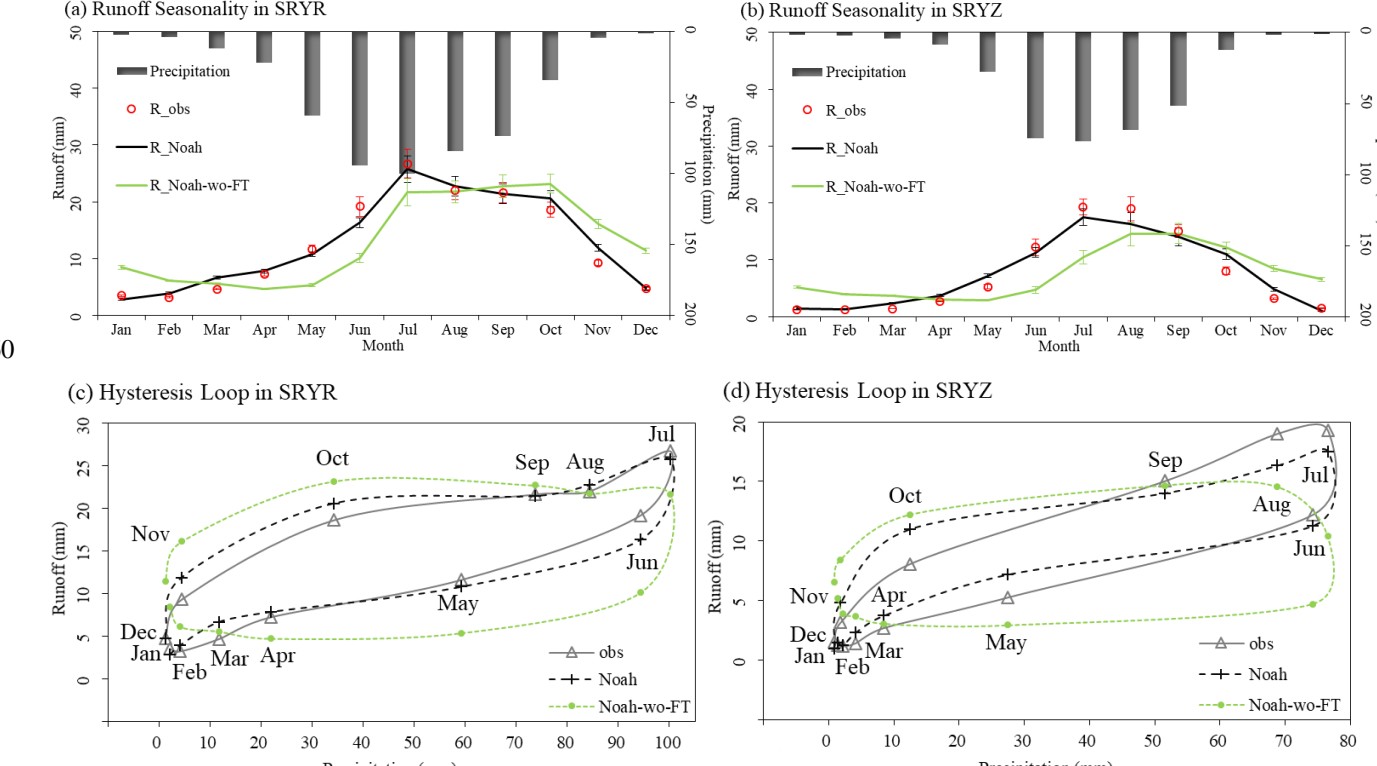

Fig. 4: Monthly mean measured (R_obs) and simulated runoff (a, b), and annual precipitation-runoff hysteresis loops (c, d) plotted from measured and simulated runoff produced by the improved Noah LSM with (Noah) and without (Noah-wo-FT) the soil freeze-thaw mechanism for the source regions of Yellow (SRYR, 1984-2009) (a, c) and Yangtze rivers (SRYZ, 1984-2005) (b, d). The figure is reproduced from Zheng et al. (2018).

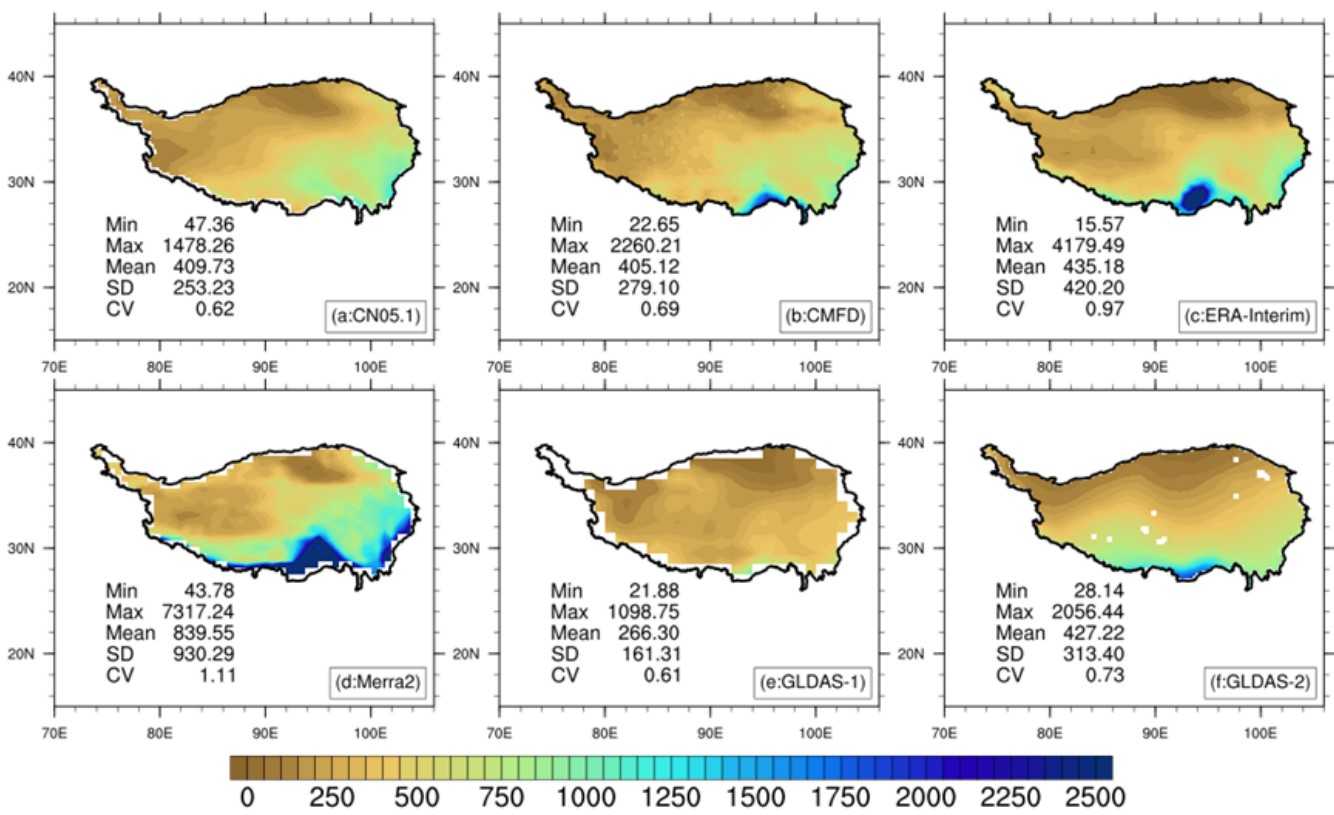

**Fig. 5:** Spatial distribution of mean annual precipitation derived from six forcing datasets for the Tibetan Plateau (1985-2014): (a) CN05.1, (b) CMFD, (c) ERA-Interim, (d) MERRA2, (e) GLDAS version1, and (f) GLDAS version 2. Statistic values, such as minimum (Min), maximum (Max), mean (Mean), standard deviation (SD), and Coefficient of Variation (CV) are also shown in each plot.

670

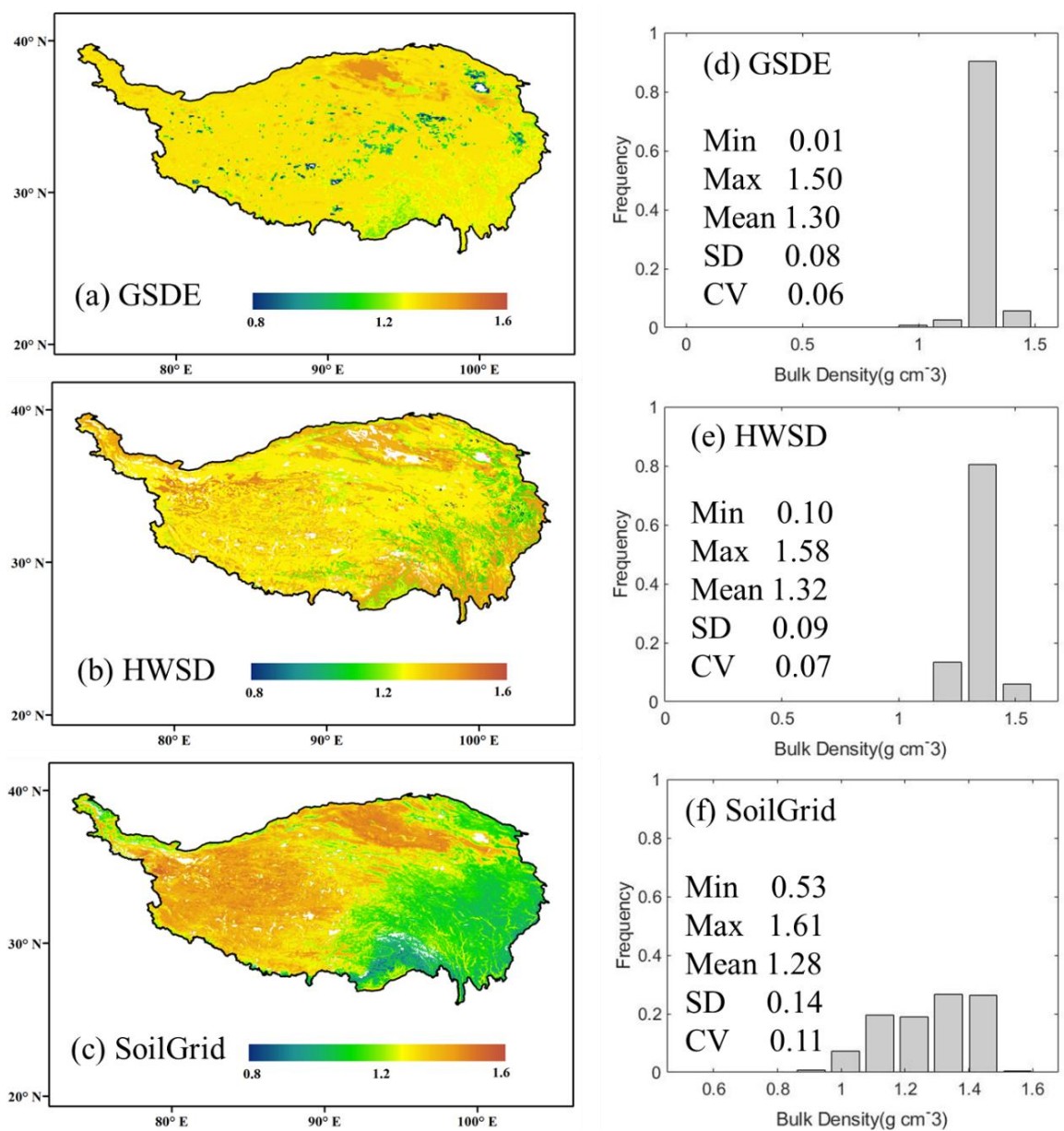

**Fig. 6: Spatial distribution of soil bulk density on the TP derived from (a) GSDE, (b) HWSD, and (c) SoilGrid, corresponding histogram are also shown in (e)-(f).**

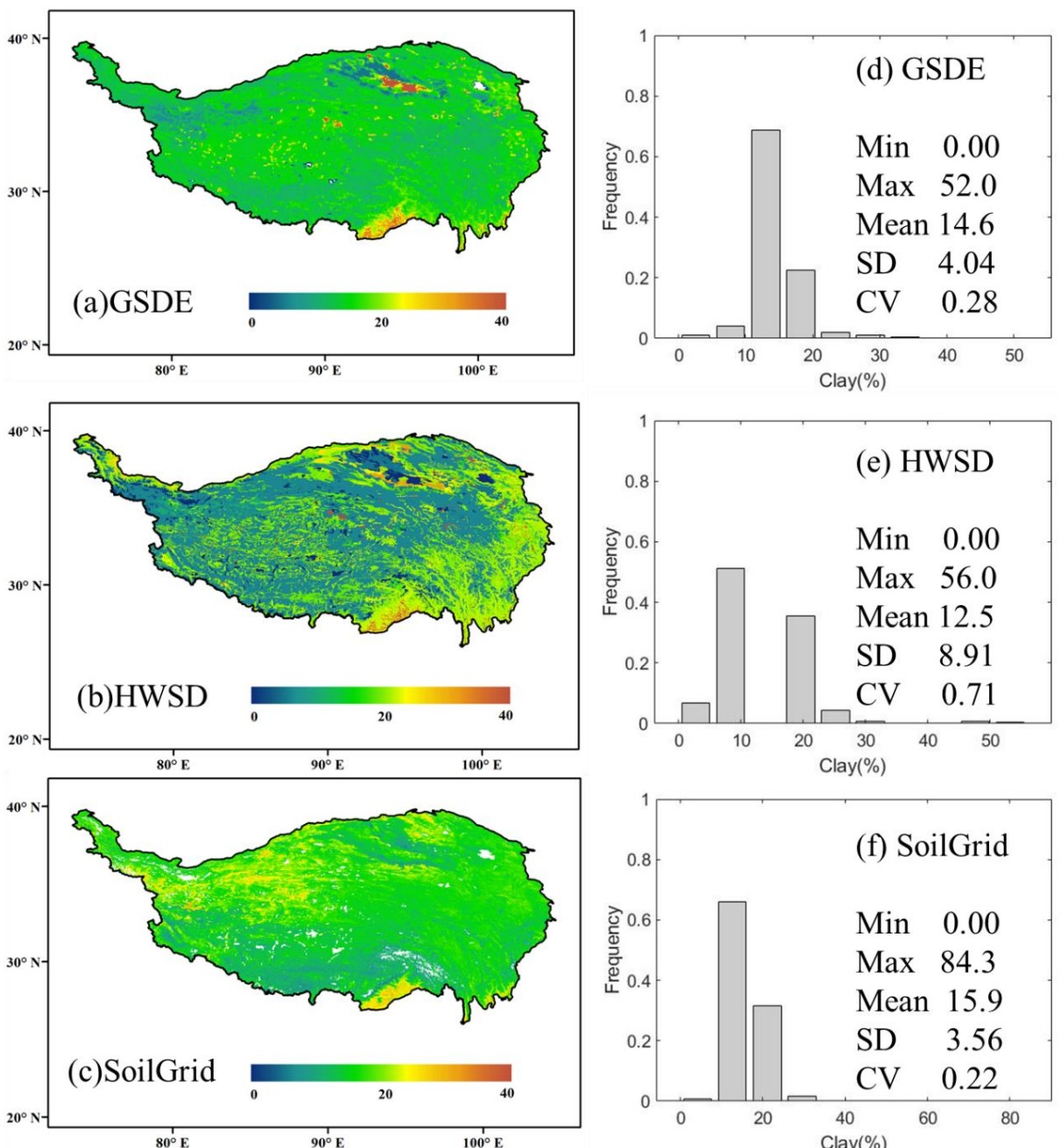

Fig. 7: Same as Fig. 6, but for soil clay content (%).

680