# Peer review of "Last Decade Progress in Understanding and Modeling the Land Surface Processes on the Tibetan Plateau"

_Hydrology and Earth System Sciences, 2020_

## Referee Comment (RC1) · Anonymous Referee #1 · 16 Aug 2020

In this study, the authors reviewed the several progress in the understanding and modelling of land surface processes on the Tibet Plateau in the last decade and summarized the major advances. This manuscript aslo provided the land surface modelling community potential directions of the further improvement of land surface modelling on TP. Overall, ,the paper is well organized and written. I only have some minor comments listed as follows:. ! The advance in the impact of Sub-grid Terrain Radiative effect over TP is missed. References are listed as follows: Gu Chunlei, Anning Huang*, et al., 2020. Effects of Sub-grid Terrain Radiative Forcing on the Ability of RegCM4.1 in the Simulation of Summer Precipitation over China. Journal of Geophysical Research: Atmospheres, DOI:10.1029/2019JD032215. Lee, W.-L., K. N. Liou, and C.-c. Wang (2013). Impact of 3-D topography on surface radiation budget over the Tibetan Plateau.

[Figure]

Theor. Appl. Climatol., 113, 95-103, doi:10.1007/s00704-012-0767-y.

2.Another missed advance is the Sub-grid orographic drag effect. references: Zhou, X., Yang, K. & Wang, Y. Implementation of a turbulent orographic form drag scheme in WRF and its application to the Tibetan Plateau. Clim Dyn 50, 2443–2455 Zhou XuÂăet al.,2019.Dynamical impact ofÂăparameterized turbulent orographic form drag onÂătheÂăsimulation ofÂăwinter precipitation overÂătheÂăwestern Tibetan Plateau.Climate Dynamics, 53:707–720.

3.I think some key lake process parameterizations including the light extinction coefficient, turbulent mixing in deep lakes, lake surface roughness length, and lake ice surface albedo need to be improved based on more field observations in the future LSM developments. This issues are necessary mentioned in perspectives on the further improvement of land surface modelling.

4.As the vegetation types are very sensitive to climate change on TP, the parameterization of dynamic vegetation should be improved further LSM development.

---

## Referee Comment (RC2) · Anonymous Referee #2 · 17 Aug 2020

This manuscript reviews the progresses in modeling land surface processes on the Tibetan Plateau (TP) in the past decade from four aspects listed in abstract. The review is relatively comprehensive. The manuscript is also well written. Regarding to the modeling land surface processes on TP, I have several comments, which have not been mentioned or mentioned but not well addressed in the manuscript. 1) In the past decade, the LSM simulations have been performed on more fine scales in comparison with previous, which were benefited from the fine resolution forcing datasets and the improved model parameterization schemes. 2) The implication of satellite observation, in particular, in the ungauged/non-observational areas, has been greatly improved our understanding the land surface processes. For example, the satellite observation provides high resolution precipitation (e.g., CMORPH, FY-x), the revolutionary of land

use/land cover, and the streamflow information etc. 3) In recent years, more in-situ meteorological stations have been installed and more field trips have also been conducted in the TP (e.g., the Second Tibetan Plateau Scientific Expedition and Research). All of them bring new information over TP which are known little in previous.

––––––––––––––––––––––––––––

---

## Referee Comment (RC3) · Anonymous Referee #3 · 4 Sep 2020

This review has great value in summarizing the progresses in land surface researches of the Tibetan Plateau and thus provides perspectives in future researches. The paper was well organized and written. Regarding to the supporting data, it is worth to review the calibration and validation site and reanalysis datasets, including soil moisture and so on. And how does them help the "observation-based" land surface researches proposed by Kun Yang et al. In addition, it may also be worthwhile to mention data-driven methods' potential in this area. Ref: 1. Deep learning process understanding data-driven Earth system science; 2. Big Earth data: A new frontier in Earth and information sciences

---

## Author Comment (AC1) · 14 Sep 2020

Response to Anonymous Referee #1

| Reviewer comment | Response |
|---|---|
| In this study, the authors reviewed the several progress in the understanding and modelling of land surface processes on the Tibet Plateau in the last decade and summarized the major advances. This manuscript also provided the land surface modelling community potential directions of the further improvement of land surface modelling on TP. Overall, the paper is well organized and written. I only have some minor comments listed as follows:. | We thank the reviewer's comments and encouragement. |
| 1. The advance in the impact of Sub-grid Terrain Radiative effect over TP is missed. References are listed as follows: Gu Chunlei, Anning Huang*, et al., 2020. Effects of Sub-grid Terrain Radiative Forcing on the Ability of RegCM4.1 in the Simulation of Summer Precipitation over China. Journal of Geophysical Research: Atmospheres, DOI:10.1029/2019JD032215. Lee, W.-L., K. N. Liou, and C.-c. Wang (2013). Impact of 3-D topography on surface radiation budget over the Tibetan Plateau. Theor. Appl. Climatol., 113, 95-103, doi:10.1007/s00704-012-0767-y. | We thank the reviewer's comments. Sub-grid terrain radiative effect is very important in land surface process. We will add this part and relative references in the revised version. |
| 2.Another missed advance is the Sub-grid orographic drag effect. references: Zhou, X., Yang, K. & Wang, Y. Implementation of a turbulent orographic form drag scheme in WRF and its application to the Tibetan Plateau. Clim Dyn 50, 2443–2455 Zhou Xu et al.,2019. Dynamical impact of parameterized turbulent orographic form drag on the simulation of winter precipitation over the western Tibetan Plateau .Climate Dynamics, 53:707–720. | We thank the reviewer's comments. We will include the sub-grid orographic drag effect in the revised version. |
| 3. I think some key lake process parameterizations including the light extinction coefficient, turbulent mixing in deep lakes, lake surface roughness length, | We thank the reviewer's suggestions. We reviewed the parameterization of vertical mixing process of lake models in current version. We will add the perspectives of lake |

| | |
|---|---|
| and lake ice surface albedo need to be improved based on more field observations in the future LSM developments. This issues are necessary mentioned in perspectives on the further improvement of land surface modelling. | model development in the revised version. |
| 4. As the vegetation types are very sensitive to climate change on TP, the parameterization of dynamic vegetation should be improved further LSM development. | We thank the reviewer's suggestions. We will consider the vegetation parameterization in the perspectives part. |

---

## Author Comment (AC2) · 14 Sep 2020

Response to Anonymous Referee #2

| Reviewer comment | Response |
|---|---|
| This manuscript reviews the progresses in modeling land surface processes on the Tibetan Plateau (TP) in the past decade from four aspects listed in abstract. The review is relatively comprehensive. The manuscript is also well written. Regarding to the modeling land surface processes on TP, I have several comments, which have not been mentioned or mentioned but not well addressed in the manuscript. | We thank the reviewer for providing a thorough and insightful review of our manuscript. |
| 1) In the past decade, the LSM simulations have been performed on more fine scales in comparison with previous, which were benefited from the fine resolution forcing datasets and the improved model parameterization schemes. | We are glad the reviewer agrees with the point that development of meteorological forcing data is important. We will introduce how the improvement of forcing data and parameterization schemes help the fine scale simulation in our revised version. |
| 2) The implication of satellite observation, in particular, in the ungauged/non-observational areas, has been greatly improved our understanding the land surface processes. For example, the satellite observation provides high resolution precipitation (e.g., CMORPH, FY-x), the revolutionary of land use/land cover, and the streamflow information etc. | We thank the reviewer's suggestions. In the revised version, we will consider the contribution of remote sensing data to improve land surface modeling and understanding. |
| 3) In recent years, more in-situ meteorological stations have been installed and more field trips have also been conducted in the TP (e.g., the Second Tibetan Plateau Scientific Expedition and Research). All of them bring new information over TP which are known little in previous. | We thank the reviewer's suggestions. We fully agree with the reviewer. As we stated in the perspectives part, "new experimental activities (e.g. the Second Scientific Expedition to the Tibetan Plateau and the Third Atmospheric Science Experiment on the Tibetan Plateau) may provide new observations" to improve land surface modeling. |

---

## Author Comment (AC3) · 14 Sep 2020

Response to Anonymous Referee #3

| Reviewer comment | Response |
|---|---|
| This review has great value in summarizing the progresses in land surface researches of the Tibetan Plateau and thus provides perspectives in future researches. The paper was well organized and written. | We thank the reviewer's comments and encouragement. |
| Regarding to the supporting data, it is worth to review the calibration and validation site and reanalysis datasets, including soil moisture and so on. And how does them help the "observation-based" land surface researches proposed by Kun Yang et al. | We thank the reviewer's suggestions. We will consider how the observation data helps to improve land surface modeling on the TP. |
| In addition, it may also be worthwhile to mention data driven methods' potential in this area. Ref: 1. Deep learning process understanding data-driven Earth system science; 2. Big Earth data: A new frontier in Earth and information sciences | We thank the reviewer's suggestions. We will add the data driven methods in the perspectives part. |

---

## Author Response (AR1)

hess-2020-348    Submitted on 03 Jul 2020

**Last Decade Progress in Understanding and Modeling the Land Surface Processes on the Tibetan Plateau**

By Hui Lu, Donghai Zheng, Kun Yang, and Fan Yang

Dear Editor and reviewers,

We would like to thank the editor and three referees for your comments and suggestions, which improve the quality of our manuscript substantially. In this document, we present the point-by-point responses to comments from you. In the marked-up manuscript version, all changes are highlighted by yellow colour.

Best Regards

Hui Lu

**Response to Editor**

**Comments:**

1) I have received three reviews for your manuscript, and they are basically positive. According to the comments, you may want to expand the review or perspectives for parameterizations related to radiation, lake and vegetation, for fine-scale data and modeling, and recent deployment of observations, and for new technology including deep learning.

**Response:**

Many thanks for your comments. In our revised version, we addressed referees' comments and suggestion by expanding the review and perspectives. Details can be found in the point-to-point responses to each referee and also in the change-tracked document.

**Response to Referee #1**

| Reviewer comment | Response |
|---|---|
| In this study, the authors reviewed the several progress in the understanding and modelling of land surface processes on the Tibet Plateau in the last decade and summarized the major advances. This manuscript also provided the land surface modelling community potential directions of the further improvement of land surface modelling on TP. Overall, the paper is well organized and written. I only have some minor comments listed as follows:. | We thank the reviewer's comments and encouragement. |
| 1. The advance in the impact of Sub-grid Terrain Radiative effect over TP is missed. References are listed as follows:
Gu Chunlei, Anning Huang*, et al., 2020. Effects of Sub-grid Terrain Radiative Forcing on the Ability of RegCM4.1 in the Simulation of Summer Precipitation over China. Journal of Geophysical Research: Atmospheres, DOI:10.1029/2019JD032215.
Lee, W.-L., K. N. Liou, and C.-c. Wang (2013). Impact of 3-D topography on surface radiation budget over the | We thank the reviewer's comments. Sub-grid terrain radiative effect is very important in land surface process. We have added a new sub-section to discuss it, from Line 179 to Line 202, as:

**"2.5 Representation of sub-grid topographic effects**

The TP is characterized by high elevation and complex terrain, which exert obvious sub-grid topographic effects on radiative and moment transfer processes. A 3D Monte Carlo photon tracing program for radiation transfer was developed by Liou et al. (2007) and applied to simulate the surface solar fluxes on the TP. It was found that the inclusion of subgrid variability can lead to a significant solar flux deviation from the conventional smoothed topography simulation. Lee et al. (2013) further improved this 3D model and applied it to investigate the topographic effect on surface radiative energy |

Tibetan Plateau. Theor. Appl. Climatol., 113, 95-103, doi:10.1007/s00704-012-0767-y.

Another missed advance is the Sub-grid orographic drag effect. references: Zhou, X., Yang, K. & Wang, Y. Implementation of a turbulent orographic form drag scheme in WRF and its application to the Tibetan Plateau. Clim Dyn 50, 2443–2455

Zhou Xu et al.,2019. Dynamical impact of parameterized turbulent orographic form drag on the simulation of winter precipitation over the western Tibetan Plateau .Climate Dynamics, 53:707–720.

budget on the TP. The results demonstrated that the TP would receive more solar flux that leads to stronger convection and enhanced snowmelt rate when the 3D topographic effects were considered. Recently, Lee et al. (2019) investigated the climate effects of the 3D radiation-topography interactions by incorporating their model into the Community Climate System Model version 4 (CCSM4). After including this topographic effect, the energy budget and air temperature simulated by the CCSM4 could be significantly improved. Gu et al. (2020) also coupled a subgrid terrain radiative forcing scheme into the Regional Climate Model Version 4.1 (RegCM4.1). They found that adopting this scheme in the RegCM4.1 could produce a better simulation of the energy budget on the TP, which thus lead to a better simulation of East Asian summer monsoon and precipitation over China.

Representing the effect of subgrid terrain variability on the moment transfer is very important for improving the simulation of wind speed, water vapour transfer, and precipitation over the TP. Wang et al. (2017) demonstrated that water vapour transfer over the South Tibetan Plateau is overestimated by current climate models. Lin et al. (2018) also found similar biases in WRF simulations and demonstrated that the complex terrain of the Himalaya can retard water vapour transfer through exerting an orographic drag on the atmosphere. Han et al. (2015) suggested an aerodynamic length of about 10 m is needed to reflect the drag, which, however, is difficult to apply such a "big number" directly to the framework of Monin-Obhukov similarity theory. Alternatively, Zhou et al. (2018; 2019) implemented a turbulent orographic form drag scheme (Bejaars et al., 2004) in the WRF model, which reduce the

| | positive biases in the surface wind simulation, leading to the reduction of water vapour transfer from the surroundings to the TP. This parameterization is also critical for the successful simulation of precipitation distribution from the low altitudes to high latitudes of the central Himalaya (Wang et al., 2020)." We also added several references to this part, including Liou et al. (2007), Lee et al. (2013; 2019), Gu et al. (2020), Wang et al. (2017), Lin et al. (2018), Han et al., (2015), Zhou et al. (2018;2019), and Wang et al. (2020). |
|---|---|
| 2. I think some key lake process parameterizations including the light extinction coefficient, turbulent mixing in deep lakes, lake surface roughness length, and lake ice surface albedo need to be improved based on more field observations in the future LSM developments. This issues are necessary mentioned in perspectives on the further improvement of land surface modelling. | We thank the reviewer's suggestions. We reviewed the parameterization of vertical mixing process of lake models in current version. We have added the lake model development in the perspectives part, from Line 306-308, as "and high elevation lakes, which are typical for the TP. For example, the parameterization of the light extinction coefficient and turbulent mixing in lakes are critical to freeze-up simulation but their values are empirically given (Huang et al., 2019)." |
| 3. As the vegetation types are very sensitive to climate change on TP, the parameterization of dynamic vegetation should be improved further LSM development. | We thank the reviewer's suggestions. We have added the vegetation parameterization in the perspectives part, from Line 308 to Line 310, as "Meanwhile, since the vegetation types on the TP have unique characteristics and are very sensitive to climate change, the parameterization of vegetation effects should be improved as demonstrated by Li et al (2019)". |

**Response to Referee #2**

| Reviewer comment | Response |
|---|---|
| This manuscript reviews the progresses in modeling land surface processes on the Tibetan Plateau (TP) in the past decade from four aspects listed in abstract. The review is relatively comprehensive. The manuscript is also well written. Regarding to the modeling land surface processes on TP, I have several comments, which have not been mentioned or mentioned but not well addressed in the manuscript. | We thank the reviewer for providing a thorough and insightful review of our manuscript. |
| 1) In the past decade, the LSM simulations have been performed on more fine scales in comparison with previous, which were benefited from the fine resolution forcing datasets and the improved model parameterization schemes. | We are glad the reviewer agrees with the point that development of meteorological forcing data is important. We emphasized this point at Line 215 to 217, as: "These improved forcing data and other supporting data generally outperform the corresponding global data set not only in accuracy but also in both spatial and temporal scales, which make a finer scale land surface simulation possible" |
| 2) The implication of satellite observation, in particular, in the ungauged/non-observational areas, has been greatly improved our understanding the land surface processes. For example, the satellite observation provides high resolution precipitation (e.g., CMORPH, FY-x), the revolutionary of land use/land | We thank the reviewer's suggestions. In the revised version, we have reviewed the contribution from remote sensing data to LSM development on the TP, in Line 269 to 272, as: "Besides the abovementioned improved forcing and soil data sets, other remote sensing or reanalysis data sets have also been widely used for land surface modeling on the TP, especially in ungauged/poor-gauged areas. They were used to drive hydrological simulation (Wang et al., 2015; Tong et al., 2014; Qi et al., 2018) or land data assimilation system |

| | |
|---|---|
| cover, and the streamflow information etc. | (Lu et al., 2012), to validate model simulation (Li et al., 2020), and to improve understanding of physical processes (Liu et al., 2018)." Some references related to satellite precipitation data validation and application on the TP were added, such as (Tong et al., 2014; Qi et al., 2018). |
| 3) In recent years, more in-situ meteorological stations have been installed and more field trips have also been conducted in the TP (e.g., the Second Tibetan Plateau Scientific Expedition and Research). All of them bring new information over TP which are known little in previous. | We thank the reviewer's suggestions. We fully agree with the reviewer. As we stated in the perspectives part, from Line 294-296, as: "new experimental activities (e.g. the Second Scientific Expedition to the Tibetan Plateau and the Third Atmospheric Science Experiment on the Tibetan Plateau) may provide new observations to improve/validate the quality of the forcing datasets." |

**Response to Referee #3**

| Reviewer comment | Response |
|---|---|
| This review has great value in summarizing the progresses in land surface researches of the Tibetan Plateau and thus provides perspectives in future researches. The paper was well organized and written. | We thank the reviewer's comments and encouragement. |
| Regarding to the supporting data, it is worth to review the calibration and validation site and reanalysis datasets, including soil moisture and so on. And how does them help the "observation-based" land surface researches proposed by Kun Yang et al. | We thank the reviewer's suggestions. We have added a paragraph to introduce the contribution of reanalysis data and ground observation to land surface study on the TP, from line 269 to Line 279, as: "Besides the abovementioned improved forcing and soil data sets, other remote sensing or reanalysis data sets have also been widely used for land surface modeling on the TP, especially in ungauged/poor-gauged areas. They were used to drive hydrological simulation (Wang et al., 2015; Tong et al., 2014; Qi et al., 2018) or land data assimilation system (Lu et al., 2012), to validate model simulation (Li et al., 2020), and to improve understanding of physical processes (Liu et al., 2018). On the other hand, current in situ observation networks also made contributions to understanding the land surface process by serving as calibration and validation sources. For example, the multiscale soil moisture and freeze–thaw monitoring network on the central TP (CTP-SMTMN) (Yang et al., 2013) and the Tibetan Plateau observatory of plateau scale soil moisture and soil temperature (Tibet-Obs) (Su et al., 2011) provide valuable data set to validate and calibrate remote sensing products and model simulations. However, |

| | the density of in situ observation networks on the TP is still very sparse. Consequently, integration of the ground observation, remote sensing products, and reanalysis data may provide a complete and reliable supporting data set for studying the land surface processes on the TP." |
|---|---|
| In addition, it may also be worthwhile to mention data driven methods' potential in this area. Ref: 1. Deep learning process understanding data-driven Earth system science; 2. Big Earth data: A new frontier in Earth and information sciences | We thank the reviewer's suggestions. We have added the data driven methods in the perspectives part, from Line 313 to 317, as:

[revised manuscript text omitted]